# Middle ear effusion, ventilation tubes and neurological development in childhood

**Jonathan Thorsen**[1], **Tine Marie Pedersen**[1,2], **Anna-Rosa Cecilie Mora-Jensen**[1,2], **Elín Bjarnadóttir**[1,2], **Søren Christensen Bager**[1,2], **Hans Bisgaard**[1†], **Jakob Stokholm**[1,2]*

1 COPSAC, Copenhagen Prospective Studies on Asthma in Childhood, Herlev and Gentofte Hospital, University of Copenhagen, Copenhagen, Denmark, 2 Department of Pediatrics, Naestved Hospital, Naestved, Denmark

☯ These authors contributed equally to this work.
† Deceased.
* stokholm@copsac.com

**Data Availability Statement:** Individual-level personally identifiable clinical data from the children participating in the cohort cannot be made

## Abstract

### Background

Otitis media with middle ear effusion (MEE) can be treated with ventilation tubes (VT) insertion, and it has been speculated that prolonged MEE in childhood can affect neurological development, which in turn may be important for later academic achievements.

### Objective

To investigate the association between middle ear effusion (MEE), treatment with ventilation tubes (VT) and childhood neurological development.

### Study design

We examined 663 children from the Copenhagen Prospective Studies on Asthma in Childhood 2010 (COPSAC$_{2010}$) unselected mother-child cohort study. Children were followed by study pediatricians with regular visits from pregnancy until 3 years of age. MEE was diagnosed using tympanometry at age 1, 2 and 3 years. Information regarding VT from age 0–3 years was obtained from national registries. We assessed age at achievement of gross motor milestones from birth, language scores at 1 and 2 years, cognitive score at 2.5 years and general development score at age 3 years using standardized quantitative tests.

### Results

Children with MEE had a lower 1-year word production vs. children with no disease: (median 2, IQR [0–6] vs. 4, IQR [1–7]; p = 0.017), and a lower 1-year word comprehension (median 36; IQR [21–63] vs. 47, IQR [27–84]; p = 0.03). Children with VT had a lower 2-5-year cognitive score vs. children with no disease; estimate -2.34; 95% CI [-4.56;-0.12]; p = 0.039. No differences were found between children with vs. without middle ear disease regarding age at achievement of gross motor milestones, word production at 2 years or the general developmental score at 3 years.

freely available, to protect the privacy of the participants and their families, in accordance with the Danish Data Protection Act and European Regulation 2016/679 of the European Parliament and of the Council (GDPR) that prohibit distribution even in pseudo-anonymized form. However, research collaborations are welcome, and data can be made available under a joint research collaboration by contacting the COPSAC Data Protection Officer (DPO), Ulrik Ralkiaer, PhD (administration@dbac.dk).

**Funding:** COPSAC is funded by private and public research funds all listed on www.copsac.com. The Lundbeck Foundation; The Danish Ministry of Health; Danish Council for Strategic Research and The Capital Region Research Foundation have provided core support for COPSAC. No pharmaceutical company was involved in the study. TMP has received funding from Oticon Foundation and GN Store Nord Foundation. The funding agencies did not have any role in design and conduct of the study; collection, management, and interpretation of the data; or preparation, review, or approval of the manuscript.

**Competing interests:** I have read the journal's policy and the authors of this manuscript have the following competing interests: JT has received a speaking fee from AstraZeneca. This does not alter our adherence to PLOS ONE policies on sharing data and materials. The authors have no other potential conflicts of interest to disclose.

**Abbreviations:** MEE, Middle Ear Effusion; VT, Ventilation Tubes; COPSAC$_{2010}$, COpenhagen Prospective Studies on Asthma in Childhood$_{2010}$; ASQ-3, Ages & Stages Questionnaires®, Third Edition; PPCA, Proportional Principal Component Analysis; PC, Principal Compone; PCA, Principal Component Analysis; IQR, Interquartile range; CI, Confidence Interval; SD, Standard deviation.

## Conclusion

Our study supports the previous findings of an association between MEE and concurrent early language development, but not later neurological endpoints up to the age of 3. As VT can be a treatment of those with symptoms of delayed development, we cannot conclude whether treatment with VT had positive or negative effects on neurodevelopment.

## Introduction

Neurological development including motor and language skills, and cognitive function is affected by genetics and environmental factors [1, 2]. Early life influences play a significant role in shaping neurological development [3] and may be important determinants of a child's subsequent academic achievements [4–7].

Otitis media is an inflammatory process within the middle ear space, presenting as either an acute infection or as middle ear effusion (MEE); a long-lasting condition with accumulation of fluid without symptoms of acute infection [8]. MEE is believed to impair children's balance and it has been speculated that prolonged MEE in early childhood can affect motor development [9, 10]. Furthermore, MEE decreases the mobility of the tympanic membrane, which can result in a mild to moderate conductive hearing loss [11] and potentially delay language development [12–14].

Prolonged MEE is often treated with ventilation tubes (VT) [15]. Treatment with VT has been shown to increase the short-term hearing ability, but with no apparent long-term effects [14].

Thus, recent studies investigating the effect of MEE/VT on language and neurodevelopmental outcomes have shown conflicting results [11, 16], warranting longitudinal prospective studies investigating such long-term effects of MEE and VT.

The aim of this study was to investigate the associations between untreated MEE, treatment with VT and children with no middle ear disease and neurological development. We used the neurological assessments: age at achievement of gross motor milestones, scores in language- and cognitive tests as well as general development of the children in the Copenhagen Prospective Studies on Asthma in Childhood$_{2010}$ (COPSAC$_{2010}$) birth cohort [17, 18].

## Materials and methods

### Study population

COPSAC$_{2010}$ is an ongoing unselected prospective mother-child cohort study of 738 pregnant women and their children recruited in pregnancy week 24 from 2008–2010 [17]. The key exclusion criteria for the pregnant women were chronic disease other than asthma or lack of fluency in Danish. Seven-hundred children were enrolled in the cohort at one week of age, excluding children with severe congenital abnormalities. For the present study, we excluded children with a neurological diagnosis and those born <37 weeks of gestation or with birth weight <2500g from all analyses. The children were followed in the COPSAC research clinics with 9 planned visits during the first three years of life. The study size was based on estimated prevalence of early asthma symptoms, as previously described [17].

### Ethics

The study was conducted in accordance with the guiding principles of the Declaration of Helsinki and was approved by the Local Ethics Committee (H-B-2008-093), and the Danish Data

Protection Agency (2015-41-3696). Both parents gave written informed consent before enrolment.

## Middle ear effusion and ventilation tubes

MEE was assessed using tympanometry performed by the research doctors in the COPSAC unit at scheduled visits at age one, two and three years. Tympanometry was performed using an Interacoustics MT10 hand-held impedance audiometer (Interacoustics, Middelfart, Denmark) with a tone of 226Hz and intensity of 85 dB SPL. A type B tympanogram (flat curve) was interpreted as MEE [19, 20], while type A or C tympanograms were considered not MEE. Information regarding VT insertions was obtained from two Danish registries, The Danish National Hospital Register [21], which contains all hospital records on VT insertions (ICD-10 diagnosis: KDCA20), and The Danish National Health Service Register (procedure code: 3009) [22], which contains information on VT insertions performed in private Ear-Nose-Throat surgeon clinics. All procedures are linked with a personal identification number of the child, leaving no missing observations [23]. For the outcome analyses, we defined a compound MEE/VT variable at each yearly timepoint as either a) Ventilation tube insertion before that date, b) Middle ear effusion (type B curve) at visit or at previous visit(s), c) No VT/MEE (type A/C curve) at visit, d) Missing data (no tympanometry measurement) at visit, in prioritized order.

## Neurological developmental assessment

**Milestones.**   The parents received a registration form at the 1-week clinical visit, with thorough instructions, based on The Denver Development Index [24] and World Health Organization milestones registration [25]. Dates of achievement of 13 predefined milestones were registered by the parents. Only the gross motor milestones: sit alone, stand with help, crawl, stand alone, walk with help and walk alone were used to analyze the association with MEE and VT because the achievement of these could be delayed by poor balance [10, 26].

**Language development.**   Language development was assessed with the Danish version of The MacArthur Bates Communicative Developmental Inventory [27]. Bilingual children were excluded from this assessment, since they may follow different temporal language acquisition patterns [28]. The assessment was performed by a web-based questionnaire filled out by parents around the child's 1-year (Words and gesticulation) and 2-year birthday (Words and sentences). From the 1-year questionnaire, we used the scores regarding word production and word comprehension. From the 2-year questionnaire, we used the word production score for the analyses.

**Cognitive development.**   Cognitive development was assessed at 2.5 years of age, using the cognitive part of the third edition of the Bayley Scales of Infant and Toddler Development [29]. During the examinations, the examiner presented a series of test materials to the child and observed the child's responses and behavior. Based on its objective performance, the child was given a composite score, which was standardized by use of a normalization material of age corrected means of 100 and a standard deviation of 15 (range 50–150). Examinations were performed by 10 trained examiners and inter-examiner consistency in performance was validated by inspection of video recordings [18].

**General neurological development.**   The parent-completed Ages & Stages Questionnaires®, Third Edition (ASQ-3) consists of 5 categories; fine motor, gross motor, personal-social, communication and problem solving skills. Parents completed the questionnaire prior to the 3 years visit, where the responses were subsequently verified. Children received a score

for each category of the ASQ-3 and we combined the 5 areas statistically and used the combined score as a general development score at the age of 3.

**Covariates.** Information regarding maternal age at birth, smoking during pregnancy, maternal asthma, delivery method, gestational age, birth weight, sex and older siblings in the household was obtained by personal interview at the 1 week visit in the COPSAC clinic. Duration of exclusive breastfeeding and age at beginning of daycare were obtained longitudinally from clinical interviews during the first year of life. Household income, maternal age and level of education were obtained from interviews when the children were 2 years of age and combined as a composite measure of the child's social circumstances defined as the z-scored first component of a PCA (explaining 55% of the variance).

## Statistics

Differences in the baseline characteristics between children with MEE, VT and with no middle ear disease were determined by Chi-square test, t-test, or Wilcoxon rank-sum test. All variables were tested for normal distribution and models were verified by visual inspection of residual plots. Trajectories were calculated as multinomial logistic regression comparing only MEE vs no disease as predictors and expressed as odds ratios (ORs). For the outcome analyses, we split the children into three groups: children with untreated MEE, children who had received VT, and children with no middle ear disease. The language data for the 1-year test were log-transformed after adding a pseudocount of 1 for the linear models to derive sex-adjusted p-values. Results were presented as medians with interquartile range (IQR) on the original scale. No transformation of the data was needed for the 2-year language data or the cognitive test. General linear models were used to analyze the association between untreated MEE, treatment with VT compared to children with no disease and their language- and cognitive development. As sex is a known determinant of neurodevelopment [30–32], also in our cohort [18], all associations were adjusted for sex.

Age of milestone achievement was analyzed using a probabilistic principal component analysis (PPCA). This model included the full dataset with missing values, assuming that the missing values were missing at random. Missing data were otherwise treated as missing observations. The 5 areas of the ASQ-3 were combined in a PCA model, which was used to extract underlying principal components (PCs) that described the systematic part of the variation across the ASQ-3 score variables in fewer uncorrelated variables. Associations between MEE and VT at age 1 year and all the study outcomes were analyzed in a combined PCA model.

A significance level of 0.05 was used in all analyses. All estimates were reported with 95% confidence intervals (CI). The data processing and analysis was conducted using R [33] version 3.6.1 with the packages pcaMethods [34] and nnet [35] and visualized using the packages ggplot2 [36] and ggforce [37].

## Results

### Baseline characteristics

A total of 700 children were included in the COPSAC$_{2010}$ cohort at birth and information on VT insertions was available for all 700 children. We excluded 37 children from the study because of neurological diagnosis, low gestational age, or low birth weight (S1 Fig), leaving 663 children. In these 663 children, tympanometry was successfully performed on 537 children at 1 year, 563 at 2 years, and 578 at 3 years (S1 Fig), showing that 51% (n = 276/537) had MEE at 1 year, 37% (n = 210/563) at 2 years and 26% (n = 155/578) at 3 years (S2 Fig). 5% (n = 36/663) of the children had VT inserted before age 1 year, 26% (n = 173/663) before 2 years and 29%

**Table 1. Baseline characteristics of the COPSAC[2010] cohort.** Children with middle ear effusion (MEE), ventilation tubes (VT) or children without middle ear disease before the age of 3. 30 children with missing assessments at 3 years were excluded from the table.

| | No middle ear disease ≤ 3 y | Middle ear effusion ≤ 3 y | P-value | Ventilation tubes ≤ 3 y | P-value |
|---|---|---|---|---|---|
| | N = 129 | n = 312 | | n = 192 | |
| Pregnancy and birth | | | | | |
| Maternal age at birth, mean (SD) | 31.59 (3.95) | 32.50 (4.31) | 0.040 | 32.44 (4.54) | 0.087 |
| Delivery, Cesarean section, n (%) | 26 (20.2) | 61 (19.6) | 0.99 | 39 (20.3) | 1.00 |
| Length in cm at birth, mean (SD) | 52.34 (2.12) | 52.08 (2.24) | 0.26 | 52.22 (1.99) | 0.62 |
| Birth weight in kg, mean (SD) | 3.63 (0.46) | 3.59 (0.49) | 0.34 | 3.60 (0.47) | 0.55 |
| Head circumference in cm at birth, mean (SD) | 35.22 (1.29) | 35.05 (1.75) | 0.30 | 35.17 (1.56) | 0.75 |
| Sex, boys, n (%) | 55 (42.6) | 164 (52.6) | 0.073 | 107 (55.7) | 0.029 |
| Ethnicity, caucasian, n (%) | 120 (93.0) | 300 (96.2) | 0.25 | 187 (97.4) | 0.11 |
| Season of birth, n (%) | | | <0.001 | | 0.21 |
| • Winter | 32 (24.8) | 114 (36.5) | | 50 (26.0) | |
| • Spring | 27 (20.9) | 91 (29.2) | | 51 (26.6) | |
| • Summer | 43 (33.3) | 45 (14.4) | | 44 (22.9) | |
| • Autumn | 27 (20.9) | 62 (19.9) | | 47 (24.5) | |
| Exposures | | | | | |
| Smoking during pregnancy, n (%) | 9 (7.0) | 24 (7.7) | 0.95 | 14 (7.3) | 1.00 |
| Older siblings, n (%) | 62 (48.1) | 178 (57.1) | 0.11 | 129 (67.2) | 0.001 |
| Duration of exclusive breastfeeding in days, median (IQR)[a] | 122 (58, 151) | 122 (55.75, 149) | 0.76 | 122 (56.50, 151) | 0.99 |
| Age at start in daycare in months, median (IQR)[a] | 10.4 (9.7, 11.9) | 10.3 (9.1, 11.8) | 0.25 | 10.1 (9.1, 11.3) | 0.023 |
| Social circumstances PCA score[b], mean (SD) | -0.04 (0.98) | 0.08 (0.99) | 0.26 | -0.09 (0.97) | 0.64 |

[a] Wilcoxon test. [b] PCA component consist of household income, maternal age, and maternal educational level at the age of 2; Higher values of the score indicate higher values for all input variables.

(n = 192/663) before 3 years (S3 Fig). Table 1 shows the baseline characteristics for children, who had either MEE or were treated with VT compared to children with no middle ear disease before the age of 3. We found more boys with VT compared to girls. Season of birth was associated with the risk of MEE, but not VT insertion. Having older siblings and the age at daycare start were associated with treatment with VT, but not with MEE.

## Trajectories of middle ear disease

The middle ear status of each child at each yearly assessment is illustrated in Fig 1. 274 children had MEE, but no VT, at the 1-year visit. At 2 years, 102 of these (37%) still had MEE, 73 (27%) had VT, 75 (27%) had no disease, and 24 (9%) had missing data. A total of 193 children had MEE, but no VT, at the 2-year visit. At 3 years, 63 (33%) of these still had MEE, 14 (7%) had VT, 95 (49%) had no disease and 21 (11%) had missing data. Compared to children with no disease, those with MEE at 1 year had increased risk of still having MEE at 2 years (OR 2.33, 95% CI [1.50;3.61], p = 0.0002), as well as increased risk of VT at 2 years (2.81, [1.70;4.64], p<0.0001). Similarly, those with MEE at 2 years had increased risk of MEE at 3 years (3.26 [1.9;5.32], p<0.0001) and VT at 3 years (5.97 [1.91;18.7], p = 0.002) although the latter transition was rare in both groups.

## Gross motor milestones

Information on achievement of at least one gross motor milestone and information regarding MEE and VT before the age of 1 year was available for 75% (n = 500/663) of the children. The overall gross motor development represented by the PC1 score of the PCA model, which

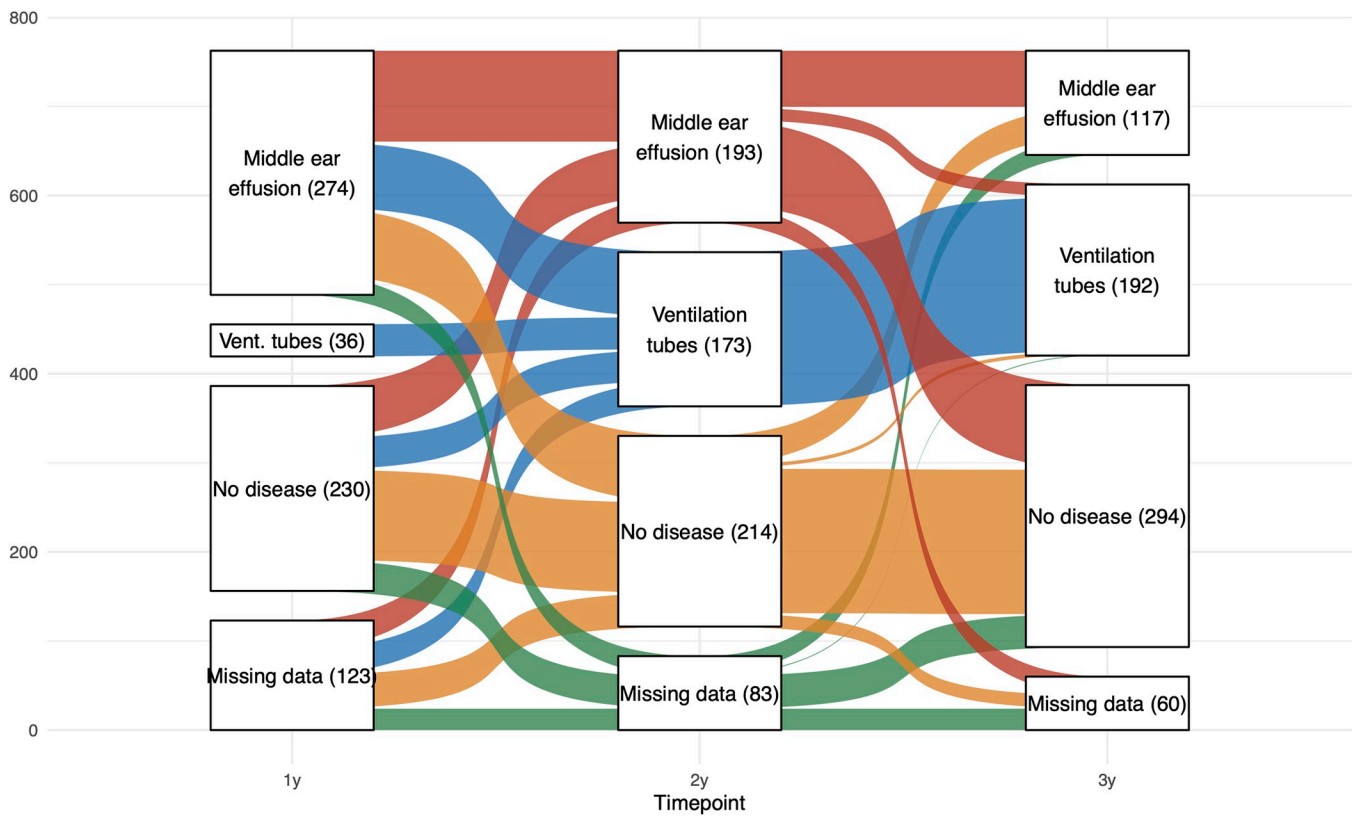

**Fig 1. Yearly trajectories of early-life chronic middle ear disease.** Middle ear effusion was assessed with yearly tympanometry measurements. Ventilation tubes was defined by start date in registries and carried forward throughout the whole period. Colors correspond to the 2-year timepoint. N = 663.

explained 63.7% of the variance in the original variables, was used to analyze the associations (S4 Fig). Neither MEE nor VT were significantly associated with age of gross motor milestone achievement compared to children without middle ear disease (Table 2 and S4 Fig).

### Language development at 1 and 2 years

Completed language assessment at 1 year and information regarding MEE and VT before the age of 1 year was obtained for 46% (n = 305/663) of the children (Fig 1). Children with MEE at age 1 year had a lower word production at the concurrent language assessment compared to children with no middle ear disease (median 2, IQR [0–6] vs. 4, IQR [1–7]), corresponding to a reduction of 24%, 95% CI [5%;39%], p = 0.02. MEE at age 1 year was also associated with a lower word comprehension compared with no middle ear disease (median 36; IQR [21–63] vs. 47; IQR [27–84]), corresponding to a reduction of 21%, 95% CI [2%;35%], p = 0.03. No associations were found between VT insertions and the 1-year language scores (Table 2).

Completed language assessment at 2 years and information on MEE at 1 or 2 years or VT insertions before age 2 years was obtained for 71% (n = 472/663) of the children. No associations were found between MEE or VT and the 2-year language scores (Table 2).

### Cognitive score

A completed cognitive test at 2.5 years and information regarding MEE at age 1 or 2 years and VT before 2.5 years was obtained for 84% (n = 558/663) of the children. VT insertion was associated with a lower composite score; adjusted estimate -2.34; 95% CI [-4.56;-0.12]; p = 0.04,

**Table 2. Associations between middle ear effusion (MEE) or treatment with ventilation tubes (VT) and neurological development.** All analyses are adjusted for sex.

| Gross Motor Milestones PPCA | | | | |
|---|---|---|---|---|
| **PC1** (higher value = faster achievement) | **N = 500** | **Estimate** | **95% CI** | **P-value** |
| MEE at 1 year | 253 | (-0.03) | [-0.37;0.30] | 0.84 |
| VT ≤ 1 year | 34 | 0.33 | [-0.34;1.01] | 0.33 |
| No VT, no MEE ≤ 1 year | 213 | reference | - | - |
| Word production at 1 year | N = 305 | Median words | IQR | P-value |
| MEE at 1 year | 159 | 2 | [0–6] | 0.017 |
| VT ≤ 1 year | 21 | 5 | [1–8] | 0.74 |
| No VT, no MEE ≤ 1 year | 125 | 4 | [1–7] | Reference |
| Word comprehension at 1 year | N = 305 | Median words | IQR | P-value |
| MEE at 1 year | 159 | 36 | [21–63] | 0.030 |
| VT ≤ 1 year | 21 | 44 | [20–76] | 1.00 |
| No VT, no MEE ≤ 1 year | 125 | 47 | [27–84] | reference |
| Word production at 2 years | N = 472 | Median words | IQR | P-value |
| MEE at 1 or 2 years | 230 | 242 | [121–356] | 0.29 |
| VT ≤ 2 years | 130 | 226 | [87–361] | 0.17 |
| No VT, no MEE ≤ 2 years | 112 | 282 | [155–370] | reference |
| **Cognitive test at 2.5 years** (higher value = faster development) | N = 558 | Estimate | 95% CI | P-value |
| MEE at 1 or 2 years | 258 | (-0.38) | [-2.43;1.66] | 0.71 |
| VT ≤ 2.5 years | 169 | (-2.34) | [-4.56;-0.12] | 0.039 |
| No VT, no MEE ≤ 2.5 years | 131 | reference | - | - |
| ASQ-3 PCA at 3 years | | | | |
| **PC1** (higher value = faster development) | N = 414 | Estimate | 95% CI | P-value |
| MEE ever (at 1, 2, or 3 years) | 211 | (-1.80) | [-4.50;0.90] | 0.19 |
| VT ≤ 3 years | 117 | (-1.55) | [-4.53;1.44] | 0.31 |
| No VT, No MEE ≤ 3 years | 86 | reference | - | - |
| Combined PCA of all neurological assessments | | | | |
| **PC1** (higher value = more developed) | N = 531 | Estimate | 95% CI | P-value |
| MEE at 1 year | 270 | (-0.15) | [-0.49;0.20] | 0.40 |
| VT ≤ 1 year | 35 | 0.42 | [-0.27;1.12] | 0.23 |
| No VT, No MEE ≤ 1 year | 226 | reference | - | - |
| **PC2** (higher value = better language and cognition) | N = 531 | Estimate | 95% CI | P-value |
| MEE at 1 year | 270 | (-0.23) | [-0.46;-0.002] | 0.048 |
| VT ≤ 1 year | 35 | (-0.04) | [-0.50;-0.42] | 0.87 |
| No VT, No MEE ≤ 1 year | 226 | reference | - | - |

PC = Principal Component. PPCA = Probabilistic Principal Component Analysis.

(Table 2). However, there were no differences between children with MEE at either 1 or 2 years and children with no middle ear disease with regards to the cognitive scores at 2.5 years.

## General development assessment (ASQ-3)

Completed ASQ-3 assessment at 3 years and information regarding MEE at age 1, 2 or 3 years or VT before 3 years of age was obtained for 62% (n = 414/663) of the children (Fig 1). The ASQ-3 PCA resulted in a PC1 score, which explained 47.5% of the variation in the data (S5 Fig). There were no differences between children with MEE at either 1, 2 or 3 years of age, VT before the age of 3 and children with no middle ear disease with regards to the general development of the child at age 3 years (Table 2).

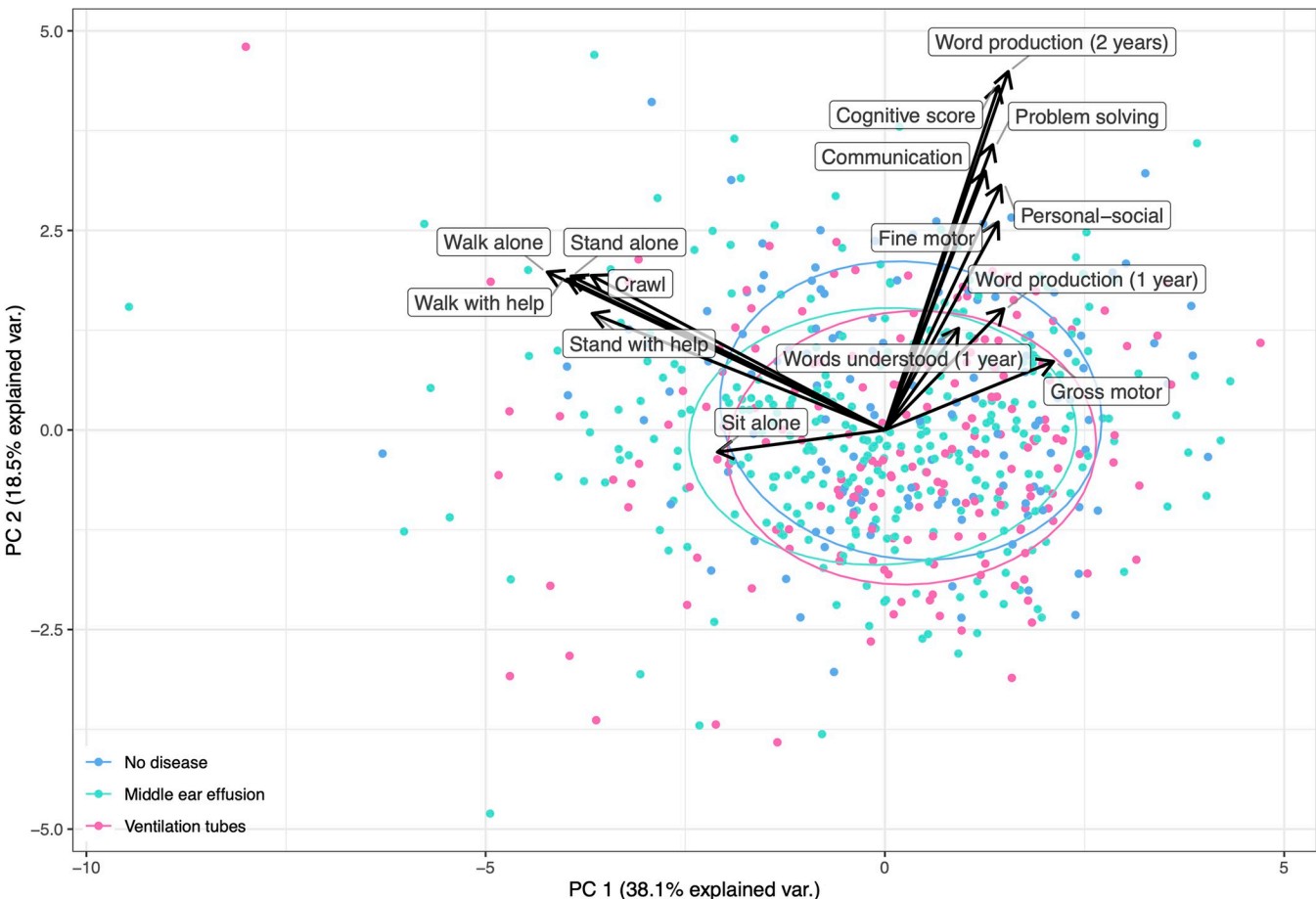

**Fig 2. PCA biplot of the neurological development scores of each child and loadings of the neurological variables.** Ellipses illustrate the scores (95% CI) of children with middle ear effusion, treatment with ventilation tubes and no disease. Gross motor milestones (Sit alone, Stand with help, Stand alone, Walk with help, Walk alone) language development at 1 and 2 years, cognitive score at 2.5 years and the ASQ-3 scores (problem solving, communication, gross motor, fine motor) at 3 years comparing children with middle ear effusion, ventilation tubes and children without middle ear disease.

## Combined PCA of neurological endpoints

To analyze effects of middle ear disease and all later neurological endpoints, we examined MEE at age 1 year and VT before 1 year in a combined PPCA model of all neurological measures. PC1 explained 38.1% of the variation in the neurological data and PC2 explained 18.5% of the variation (Fig 2). MEE at 1 year was significantly associated with a lower PC2-value; adjusted estimate -0.23; 95% CI [-0.46;-0.002]; p = 0.048; but was not associated with PC1 and no significant associations were found for VT before 1 year of age.

## Discussion

### Primary findings

In an unselected prospective birth cohort, we assessed MEE and VT longitudinally in the first three years of life in 663 children. While most untreated MEE remitted spontaneously, a significantly increased risk of persistence or VT insertion in the following year was found both for the 1- and 2-year timepoint. We found slightly lower language scores at age 1-year among children with concurrent MEE compared to children without middle ear disease, but no differences in the 2-year language scores and no associations between VT and language scores.

Children who received VT had a reduced cognitive score at 2.5 years, but without associations among children who had MEE. No differences were found between children with MEE or VT and those without disease for gross motor milestone achievement or for the general development ASQ-3 score at 3 years of age.

## Strengths and limitations

The children in the COPSAC$_{2010}$ cohort have been followed prospectively since birth with several neurological assessments to evaluate the development of the children. We have both early and late measures of the motor development. We have evaluated the language development both at age 1 and 2 years and as part of the ASQ-3 at 3 years. Furthermore, we evaluated the cognitive development of the children by the objective cognitive test in the research clinic. This provides a unique collection of standardized neurological developmental tests performed prospectively in a large cohort of children.

Not all children completed all tests (S1 Fig), which could limit the statistical power. Missing information on specific milestones was handled by the PPCA model. A single missing milestone would otherwise have excluded the child in a traditional PCA model.

MEE was diagnosed by the tympanometry measurement, which is an objective, well recognized test and easy to perform. The limitation is the nature of an annual spot measurement in the COPSAC clinic, as children may have had periods of MEE between the examinations. However, most children are affected by MEE over longer periods of time, making the yearly measurements an acceptable measure of MEE burden. Season of birth was associated with MEE but not with VT. This could be due to the study design with annual measurements, since children were assessed in the same season around their birthday every year (Table 1).

The Danish registries strengthen the study as they contain information regarding all VT procedures and contacts with the health care system linked to a personal identification number. Therefore, we have information regarding VT procedures on all 700 children in the cohort. The registries do not contain information on the VT indication (middle ear effusion, recurrent acute otitis media or other reason), or for how long the VT was in place, which may limit our interpretation.

The study is limited by its observational design. Since treatment with ventilation tubes was performed by indication, we cannot know if any observed differences between groups are due to disease history, treatment, or both. Similarly, we cannot observe the counterfactual non-treatment outcomes in the children who were treated with VTs. Only a randomized trial can evaluate the efficacy of VT insertion with respect to later developmental endpoints.

## Interpretation

We found that approximately half of 1-year old children, reducing to one quarter of 3-year-old children, have MEE as assessed by tympanometry. Furthermore, the risk of subsequent MEE was increased in those children who previously had MEE.

In our study, MEE but not VT was associated with language scores at 1 but not 2 years. This could indicate that the VT insertion is beneficial for the short-term hearing ability; however, at 2 years of age, we found no difference between the children with MEE or VT and children without disease regarding their language development. The Australian Raine cohort study (n = 1344) found only a very slightly reduced rate of vocabulary growth at 10 years in children who had bilateral OM at 6 years [38]. These observations are supported by previous literature describing MEE to cause a mild hearing impairment, which often resolves spontaneously [14].

We found an increased rate of VT in children with siblings, and a non-significant trend towards the same for MEE. This could be due to increased risk of infections in children with

siblings [39], or due to an effect on the airway microbiota [40], which may in turn influence infection susceptibility [41].

We found that the cognitive test at the age of 2.5 years was associated with VT but not MEE. This could indicate that treatment with VT did not have beneficial effects on the cognitive development of the children or simply indicate confounding by indication; the children treated with VT were those, who had the most symptoms of delayed development, which is consistent with current guidelines [15].

We observed no associations between early middle ear disease and neither the gross motor milestones nor the general development measured by ASQ-3 [42]. The combined PCA analysis of all recorded neurological measures showed that MEE at 1 year was significantly association with PC2 (mainly driven by lower language and cognitive scores) (Fig 2). No association was found for VT in the combined PCA analysis, which could be because only few children were treated with VT before 1 year of age. The result of this combined neurological PCA supports the individual endpoint analyses.

The literature shows conflicting results regarding the beneficial effects of treatment with VT on neurodevelopmental outcomes [11, 16, 43–45]. In the American study by Paradise et al., children with MEE were randomized to treatment either promptly with VT or after 9 months if MEE persisted, and they were followed until the age of 9–11 years with developmental evaluations. They found no difference between children in the two groups regarding cognitive, language, speech, or psychosocial development [46, 47]. In the Australian Raine cohort, recurrent OM at 3 years was associated with behavior at 10 years of age [48]. Developmental outcomes may vary between countries. It is known that Danish children generally score lower in early language tests, probably caused by the nature of the Danish sound structure [49]. Other factors potentially accounting for discrepancies between studies could be cultural differences such as age of daycare attendance [49].

Our study supports the previous findings of an effect of MEE on concurrent early language development. Except for the 1-year language scores, the children with VT had lower estimates compared to children with MEE or children without middle ear disease. We cannot conclude whether treatment with VT had any beneficial neurological effects, but we demonstrated no long-term differences between treated and untreated children.

In our cohort 29% of the children received VT before the age of 3 [50], which is a very high prevalence for a treatment with no solid evidence of long-term beneficial effects. Well-designed randomized controlled trials to gauge the efficacy of VTs for improving neurological development are warranted.

## Conclusion

MEE and VT insertion were very common in this cohort of 663 unselected Danish children. MEE displayed a high degree of recurrence and risk of subsequent VT insertion. Children with MEE at 1 year had slightly lower language scores at 1-year of age, but they were not affected in any other neurological endpoints to age 3 years. We did not find any major differences in the neurological development between children treated with VT and children without middle ear disease. In summary, children with middle ear disease in the first three years of life did not have materially delayed neurologic development by age 3.

## Supporting information

**S1 Fig. Flowchart and participation in each study assessment and outcome analysis.** (TIF)

**S2 Fig. Prevalence of middle ear effusion measured by tympanometry at 1, 2 and 3 years of age in the COPSAC$_{2010}$ cohort.**
(TIF)

**S3 Fig. Kaplan-Meier curve illustrating time to first ventilation tube insertion in the COP-SAC$_{2010}$ cohort.**
(TIF)

**S4 Fig. PPCA biplot of the gross motor milestone scores of each child and loadings of the neurological variables.** Ellipses illustrate the scores (95% CI) of children with middle ear effusion, treatment with ventilation tubes and no disease in the first year of life. The figure shows that there is no difference between children with middle ear effusion, ventilation tubes and no disease in the age of achieving the gross motor milestones. High PC1 scores can be interpreted as lower age at achievement of alle the gross motor milestones, especially Stand with help, Stand alone, Walk with help, Walk alone.
(TIF)

**S5 Fig. PCA biplot of the ASQ-3 scores of each child and loadings of the neurological variables.** Ellipses illustrate the scores (95% CI) of children with middle ear effusion, treatment with ventilation tubes and children with no disease. The ASQ-3 consists of 5 categories; fine motor development, gross motor development, personal-social skills, communication and problem solving. When the scores are analyzed together as one measure of the child's development it results in a PC1 score, which explains 47.5% of the variation. Higher values on PC1 equates to higher scores in all categories, especially Fine motor and Problem solving, which can be interpreted as children with high PC1 scores are further in their development than those with low PC1 scores.
(TIF)

## Acknowledgments

We express our deepest gratitude to the children and families of the COPSAC$_{2010}$ cohort study for all their support and commitment. We acknowledge and appreciate the unique efforts of the COPSAC research team.

## Governance

We are aware of and comply with recognized codes of good research practice, including the Danish Code of Conduct for Research Integrity. We comply with national and international rules on the safety and rights of patients and healthy subjects, including Good Clinical Practice (GCP) as defined in the EU's Directive on Good Clinical Practice, the International Conference on Harmonisation's (ICH) good clinical practice guidelines and the Helsinki Declaration. We follow national and international rules on the processing of personal data, including the Danish Act on Processing of Personal Data and the practice of the Danish Data Inspectorate.

## Author Contributions

**Conceptualization:** Jonathan Thorsen, Tine Marie Pedersen, Hans Bisgaard, Jakob Stokholm.

**Data curation:** Jonathan Thorsen, Tine Marie Pedersen, Anna-Rosa Cecilie Mora-Jensen, Elín Bjarnadóttir.

**Formal analysis:** Jonathan Thorsen, Tine Marie Pedersen, Jakob Stokholm.

**Funding acquisition:** Tine Marie Pedersen, Hans Bisgaard, Jakob Stokholm.

**Investigation:** Jonathan Thorsen, Tine Marie Pedersen, Anna-Rosa Cecilie Mora-Jensen, Elín Bjarnadóttir, Søren Christensen Bager, Hans Bisgaard, Jakob Stokholm.

**Methodology:** Tine Marie Pedersen, Elín Bjarnadóttir, Hans Bisgaard, Jakob Stokholm.

**Project administration:** Jonathan Thorsen, Tine Marie Pedersen, Hans Bisgaard, Jakob Stokholm.

**Supervision:** Hans Bisgaard, Jakob Stokholm.

**Validation:** Jonathan Thorsen, Jakob Stokholm.

**Visualization:** Jonathan Thorsen.

**Writing – original draft:** Jonathan Thorsen, Tine Marie Pedersen, Jakob Stokholm.

**Writing – review & editing:** Jonathan Thorsen, Tine Marie Pedersen, Anna-Rosa Cecilie Mora-Jensen, Elín Bjarnadóttir, Søren Christensen Bager, Hans Bisgaard, Jakob Stokholm.

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
