## [Decision Letter · Decision Letter 0]

25 Nov 2022

PONE-D-22-25558Middle ear effusion, ventilation tubes and neurological development in childhoodPLOS ONE

Dear Dr. Stokholm,

Thank you for submitting your manuscript to PLOS ONE. After careful consideration, we feel that it has merit but does not fully meet PLOS ONE’s publication criteria as it currently stands. Therefore, we invite you to submit a revised version of the manuscript that addresses the points raised during the review process.

Although the study is interesting, there is a major concern regarding data availability. This is one of the requirements of Plos One (please see "Additional Editor Comments" section below). 

We look forward to receiving your revised manuscript.

Kind regards,

Rafael da Costa Monsanto, M.D.

Academic Editor

PLOS ONE

Journal Requirements:

“I have read the journal's policy and the authors of this manuscript have the following competing interests: JT has received a speaking fee from AstraZeneca”

Additional Editor Comments:

One major concern the editorial board has is regarding the following statement:

"We note your Data Availability statement: "Individual-level clinical data from the children participating in the cohort cannot be made freely available, to protect the privacy of the participants and their families. However, research collaborations are welcome and data can be made available under a joint research collaboration by contacting the corresponding author."

PLOS journals require authors to make all data necessary to replicate their study’s findings publicly available without restriction at the time of publication. When specific legal or ethical restrictions prohibit public sharing of a data set, authors must indicate how others may obtain access to the data.

Please address the following:

a) At this time disclose the reason for this data restriction and provide a point of contact, preferably an email, for the ethical body mandating the restriction of this data. Please note that this point of contact cannot be an author of this study.

b) If there are no restrictions on this dataset, please provide your data by uploading it as a Supporting Information file or depositing it in a stable repository.

We hope to hear from you soon.

Reviewers' comments:

Reviewer's Responses to Questions

**Comments to the Author**

1. Is the manuscript technically sound, and do the data support the conclusions?

Reviewer #1: Yes

Reviewer #2: Partly

2. Has the statistical analysis been performed appropriately and rigorously? 

Reviewer #1: Yes

Reviewer #2: N/A

3. Have the authors made all data underlying the findings in their manuscript fully available?

Reviewer #1: Yes

Reviewer #2: Yes

4. Is the manuscript presented in an intelligible fashion and written in standard English?

Reviewer #1: Yes

Reviewer #2: Yes

5. Review Comments to the Author

Reviewer #1: The issue of the effects of acute otitis media on motor and language development still needs to be addressed. Therefore, cohorts with long periods are studies that can help to understand the real causal involvement of the inflammatory processes of the middle ear and the treatment with the placement of the ventilation tube.

I congratulate authors for this manuscript. It is very well written, with a huge sample providing relevant data due to its 3-year follow-up, apparently with little missing data for different endpoint analyzed.

However, I would like to point out one aspect that need clarification.

- In the results in relation to the Baseline Characteristics, it was confusing because it is very clear that of the 700 children followed, only 37 were excluded in relation to the placement of Ventilation tubes, however, when reporting the % of ESM diagnosis with tympanometry, it is reported that 51% (n= 276/537) had MEE at 1 year, 37% (n=210/563) at 2 years and 26% (n=155/578) at 3 years. Are these numbers 537, 563, 578 for the different years and not 663 due to missing data? Please make this clearer in the text.

Reviewer #2: Dear authors:

Although the manuscript is generally interesting, I have a few concerns:

ABSTRACT

Should be adjusted as per the suggestions below.

INTRODUCTION

Although your manuscript is generally interesting, it was unclear how your work adds to the already existing literature in this regard. The introduction is highly summarized and does not address the literature gap that the study aims to fill. I would strongly recommend that the authors reformulate their introduction to address this.

METHODS

The exclusion criteria should be better delineated:

A) In ‘Study Population’ the phrase: - “The key exclusion criteria were chronic disease other than asthma or lack of fluency in Danish”. Does it refer to the mother or the child?

B) In ‘Middle ear effusion and ventilation tubes’ the phases: - “while type A or C tympanograms were considered normal” and “c) No disease (type A/C curve)”, should be better described, as a type C tympanogram, especially in a child, should never be considered normal.

C) In ‘Language development’ it is described that: “Bilingual children were excluded from this assessment”. It is important to explain why and also put in the exclusion criteria.

D) In ‘Covariates’ the authors cite that they excluded “37 children from the study because of neurological diagnosis, low gestational age, or low birth weight”. it should be in the exclusion criteria

I believe that the data on the hearing thresholds of these children would be very enriching, as the degree of hearing loss greatly influences language acquisition.

RESULTS

It would be important to evaluate information regarding: “Gross motor milestones and Cognitive score”, because the data are accounted for with 700 children, but in the results, it was mentioned that some were excluded.

Tables: Tables are confusing and are not consistent with usual scientific standards.

DISCUSSION

Discussion is largely based on a repetition of the results. I missed a more in-depth critical analysis of their results, especially on how your results compares with those of previous studies.

6. PLOS authors have the option to publish the peer review history of their article (what does this mean?). If published, this will include your full peer review and any attached files.

Reviewer #1: No

Reviewer #2: No

---

## [Decision Letter · Decision Letter 1]

22 Dec 2022

Middle ear effusion, ventilation tubes and neurological development in childhood

PONE-D-22-25558R1

Dear Dr. Stokholm,

We’re pleased to inform you that your manuscript has been judged scientifically suitable for publication and will be formally accepted for publication once it meets all outstanding technical requirements.

Kind regards,

Rafael da Costa Monsanto, M.D.

Academic Editor

PLOS ONE

Additional Editor Comments (optional):

Reviewers' comments:

Reviewer's Responses to Questions

**Comments to the Author**

1. If the authors have adequately addressed your comments raised in a previous round of review and you feel that this manuscript is now acceptable for publication, you may indicate that here to bypass the “Comments to the Author” section, enter your conflict of interest statement in the “Confidential to Editor” section, and submit your "Accept" recommendation.

Reviewer #1: All comments have been addressed

Reviewer #2: All comments have been addressed

2. Is the manuscript technically sound, and do the data support the conclusions?

Reviewer #1: Yes

Reviewer #2: Yes

3. Has the statistical analysis been performed appropriately and rigorously? 

Reviewer #1: Yes

Reviewer #2: Yes

4. Have the authors made all data underlying the findings in their manuscript fully available?

Reviewer #1: No

Reviewer #2: Yes

5. Is the manuscript presented in an intelligible fashion and written in standard English?

Reviewer #1: Yes

Reviewer #2: Yes

6. Review Comments to the Author

Reviewer #1: Thank you for thoroughly addressing the comments. However, the study data underlying the findings were not made available because the authors explained that they are following the country's privacy regulation that does not allow the individual disclosure of the children involved.

Reviewer #2: (No Response)

7. PLOS authors have the option to publish the peer review history of their article (what does this mean?). If published, this will include your full peer review and any attached files.

Reviewer #1: No

Reviewer #2: No

---

## [Editor Report · Acceptance letter]

4 Jan 2023

PONE-D-22-25558R1 

Middle ear effusion, ventilation tubes and neurological development in childhood 

Dear Dr. Stokholm:

I'm pleased to inform you that your manuscript has been deemed suitable for publication in PLOS ONE. Congratulations! Your manuscript is now with our production department. 

Kind regards, 

on behalf of

Dr. Rafael da Costa Monsanto 

Academic Editor

PLOS ONE